# Factors Affecting Perceived Effectiveness of Government Response towards COVID-19 Vaccination in Occidental Mindoro, Philippines

**DOI:** 10.3390/healthcare10081483

**Published:** 2022-08-07

**Authors:** Yung-Tsan Jou, Klint Allen Mariñas, Charmine Sheena Saflor, Michael Nayat Young, Yogi Tri Prasetyo, Satria Fadil Persada

**Affiliations:** 1Department of Industrial and Systems Engineering, Chung Yuan Christian University, Taoyuan 320, Taiwan; 2School of Industrial Engineering and Engineering Management, Mapua University, Manila 1002, Philippines; 3Department of Industrial Engineering, Occidental Mindoro State College, San Jose 5100, Philippines; 4Entrepreneurship Department, Binus Business School Undergraduate Program, Bina Nusantara University, Jakarta 11480, Indonesia

**Keywords:** COVID-19, vaccination, perceived effectiveness, government response

## Abstract

The COVID-19 pandemic has caused several developing countries to fall behind on vaccination at the onset of the pandemic, thus affecting the mobility of easing restrictions and lowering virus transmission. The current study integrated the Protection Motivation Theory (PMT) and extended the Theory of Planned Behavior (TPB) to evaluate factors affecting the perceived effectiveness of government response towards COVID-19 vaccination in Occidental Mindoro. A total of 400 respondents from the municipalities of Occidental Mindoro answered the online questionnaires, which contained 61 questions. This study outlined the relationship between the dependent and independent variables using structural equation modeling (SEM). The results indicated that knowledge of COVID-19 vaccination had significant direct effects on its perceived severity. Subjective standards had significant adverse effects on willingness to follow. In addition, perceived behavioral control was discovered to impact willingness to follow positively. It also showed that perceived government response was significantly affected by adaptive behavior and actual behavior regarding the perceived government response. Meanwhile, it was found that the perceived government response had significant effects on perceived effectiveness. The current study is one of the first to study the factors that affect the perceived effectiveness of government response toward COVID- 19 vaccination.

## 1. Introduction

COVID-19 first appeared in Wuhan, China, in December 2019, and was declared as a “pandemic” on 11 March 2020. The Philippines is one of the Western Pacific countries most badly hit by COVID-19. The first case of COVID-19 was confirmed by the Philippine Department of Health (DOH) on 20 January 2020 [1,2]. Since then, more than 180 million cases have been recorded worldwide and around 3 million deaths as of 6 July 2021 [3]. The Philippines recorded around 2.8 million cases and more than 51,000 deaths as of December 2021 and remains one of the most affected countries in the Western Pacific region [4]. Moreover, healthcare systems have been disrupted throughout the pandemic due to emerging COVID-19 variants such as delta and omicron due to the easing of restrictions and the lagging of vaccination.

The lagging of vaccination in the country affects the mobility of easing restrictions and lowering virus transmission. The Philippines administered more than 60 million first doses and over 46 million second doses, with 104 million doses of COVID-19 vaccines as of December 2021 [5]. The Philippine government is still far from reaching the herd immunity rate of 60% to 70% of the total population. One possible reason is the current perception of the effectiveness of the vaccine. Similarly, in the US, a significant number of the population (25 to 35%) question the safety and effectiveness of the COVID-19 vaccines [6,7]. Several researchers have studied the effects of COVID-19 on emotion, education, response, transportation, and economy [8,9,10,11,12]; however, a limited study investigated the perceived effectiveness of the government response toward COVID-19 vaccination.

The study investigated the factors that affect the perceived effectiveness of government response towards COVID-19 vaccination at Occidental Mindoro, Philippines, applying the Theory of Planned Behavior (TPB) and the Protection Motivation Theory (PMT). The study might be one of the first to analyze the perceived effectiveness of government response to COVID-19 vaccinations and to possibly provide the information to the government agencies in enhancing public trust in their government response to COVID-19 vaccinations.

### Theoretical Framework

The study proposed 13 hypotheses, as shown in the hypothesized SEM model (Figure 1), which served as the study’s theoretical research framework. We used the PMT and TPB to investigate the causal relationships between identified factors and latent variables, unlike a prior study that solely used PMT linked to COVID-19 [13]. Similarly to Yogi et al. (2020), the current study integrated the PMT and TPB in evaluating the factors affecting the perceived effectiveness of the government response towards COVID-19 vaccination in Occidental Mindoro, Philippines [14].

The Protection Motivation Theory occurs when a person is mainly driven to undertake a defensive action due to an unexpected event [15]. Perceived vulnerability, also called perceived probability, indicates a person’s assumptions about the possibility of a health risk or the probability of having a disease. Weinstein (2000) indicated that people progress to various phases in understanding their vulnerability. It commonly starts from being unaware to having awareness and finally recognizing the risk they might experience. Awareness is a critical factor in preventing any harm from happening. It might benefit individuals in understanding certain events; thus, it prohibits the occurrence of the risk [16]. Thus, the researchers hypothesized the following:

**Hypothesis** **(H1).***There is a significant relationship between knowledge of COVID-19 vaccination and perceived vulnerability*.

**Hypothesis** **(H2).***There is a significant relationship between knowledge of COVID-19 vaccination and perceived severity*.

Individuals with varying global and personal perceptions (severity and vulnerability) of COVID-19 may exhibit varying behavioral responses to COVID-19. According to several studies on perceived vulnerability and severity, the disease appears to promote engagement and compliance with COVID-19 preventative behavior [17,18,19]. Thus, the researchers hypothesized the following:

**Hypothesis** **(H3).***There is a significant relationship between perceived vulnerability and perceived behavioral control*.

**Hypothesis** **(H4).***There is a significant relationship between perceived severity and perceived behavioral control*.

In accordance with the Theory of Planned Behavior, a person’s beliefs about whether or not other people should engage in a behavior determine whether they should engage in it. An individual’s emotions become intense when he or she is close to being unwell (HBM). People typically consider the medical and social consequences when measuring severity. It does not address environmental or economic factors that could influence a person’s decision to engage in a behavior and whether to prohibit or promote it [20]. Thus, the researchers hypothesized the following:

**Hypothesis** **(H5).***There is a significant relationship between perceived vulnerability and subjective standard*.

**Hypothesis** **(H6).***There is a significant relationship between perceived severity and subjective standard*.

Perceived behavioral control has a correlation to the factors derived from the theory of planned behavior [21]. Self-efficacy is a component shared by TPB and PMT; both theories have a similar notion of perceived behavioral control [22]. It emphasizes a person’s skills and competencies in handling a task or planning [23]. Moreover, it also exhibits that self-efficacy considerably influences a person’s capability in performing task behavior [24]. Therefore, the researchers hypothesized that:

**Hypothesis** **(H7).***There is a significant relationship between perceived behavioral control and willingness to follow*.

According to Fan et al. (2021), the subjective norm of COVID-19 vaccination is how an individual perceives others’ opinions toward COVID-19 vaccination [25]. Past research indicated that there is a significant relationship between subjective standards and organizational conformity [26,27,28]. In contrast, Armitage and Conner (2001) indicated that subjective standards commonly show a poor indicator of behavioral intentions due to personal perception and environmental factors [29]. Hence, we hypothesized that:

**Hypothesis** **(H8).***There is a significant relationship between subjective standards and willingness to follow*.

As stated in the TPB, the prototype willingness model is a systematic process that influences achieving behavioral indication. Behavior is susceptible to instantaneous changes in the environment, is impulsive, and is significantly affected by the perception of likelihood of a behavioral model, whereas when individuals find themselves in situations that encourage certain behaviors, especially risk-taking behaviors such as smoking, it is not their preconceived intentions that determine their actions but their eagerness to associate with the behaviors such as an interest in an opportunity. The extent to which individuals regard themselves as similar to the prototype person who engages in the actions in question determines their willingness [30]. Thus, the researchers hypothesized the following:

**Hypothesis** **(H9).***There is a significant relationship between willingness to follow and adaptive behavior*.

**Hypothesis** **(H10).***There is a significant relationship between willingness to follow and actual behavior*.

COVID-19 was one of the most challenging crises the world has faced in the last century. The Philippines had the world’s longest lockdown, in which entire provinces and cities were placed under lockdown, transportation was prohibited, and masks and social distance were rigidly enforced. Punitive action was taken in response to violations. Maintaining order and adhering to all health procedures are the responsibility of the police and military. The government’s response has been labeled as “draconian”, “militarized”, or “police-centric” by some observers and academics [29]. According to Owens (2000), even though people are aware of the situation and desire to help, they may see their actions as insignificant on a bigger scale. As a result, in addition to retaining prosocial beliefs, socially responsible behavior requires the belief that one’s activities will have an impact [30]. Thus, the researchers hypothesized the following:

**Hypothesis** **(H11).***There is a significant relationship between actual behavior and perceived government response*.

**Hypothesis** **(H12).***There is a significant relationship between adaptive behavior and perceived government response*.

**Hypothesis** **(H13).***There is a significant relationship between government response and perceived effectiveness*.

## 2. Methodology

### 2.1. Participants

The study utilized a cross-sectional design due to the social distancing measures and restricted movement and lockdowns; data were collected online, using Google Forms [14]. A link to the survey was distributed to participants through Messenger from 2 October to 30 November 2021. Informed consent was obtained from all subjects involved in the study before conducting the survey. The total population of 11 municipalities in the province of Occidental Mindoro is 525,354 [31]. To determine the sample size and the representativeness of the population, convenience sampling was used and Slovin’s formula was utilized. A total of 400 respondents from the different municipalities of Occidental Mindoro between 18 to 70 years old answered the online questionnaire, which contained 61 questions.

The result of the questionnaire showed that, among the 400 participants, 56.8% were female and 43.2% were male. About 80.5% were between 18–29 years of age, 6.3% were between 30–39, 6.8% were between 40 and 49 years, 5.2% were between 50–59, 1% were between 60–69, and 0.2% were 70 and above. Approximately 2.5% of the respondents were elementary graduates, 10.5% were high school graduates, 42% were senior high school graduates, 5.8% were technical/vocational graduates, 29.8% were baccalaureate/college graduates, 4% were post-baccalaureate graduates, 3.5% were special education undergraduate graduates, 0.7% were special education graduates, and 1.2% were no grade completed. In addition, most of the participants were from the municipalities of San Jose with 35.2%, Rizal with 12.5%, Magsaysay with 12.5%, Sta. Cruz with 9.2%, Sablayan with 6.2%, Looc with 4.8%, Abra de Ilog with 4.5%, Paluan with 3.7%, Mamburao with also 3.7%, Lubang with 4%, and, lastly, Calintaan with 3.5%. Additionally, most have a monthly salary/allowance of less than PHP 15,000 (77.3%). About 16% of the respondents have a monthly salary/allowance of PHP 15,000–PHP 30,000, 5% of them have a monthly salary/allowance of PHP 30,000–PHP 45,000, 1% of them have a salary of PHP 45,001–PHP 60,000, and 0.7% have a salary of PHP 60,001–PHP 75,000, as shown in Table 1.

### 2.2. Questionnaire

A self-administered questionnaire from different related studies was developed to evaluate the factors affecting the perceived effectiveness of government response towards COVID-19 vaccination in Occidental Mindoro. The questionnaire consisted of 61 questions with 11 sections: (1) demographic profile (gender, age, educational background, municipality, monthly salary, and health insurance; it was required to determine if the subjects in the given research constituted a representative sample of the target population, for much better results for this study); (2) knowledge of COVID-19 vaccination; (3) perceived vulnerability to disease; (4) perceived severity to disease; (5) perceptions towards government response; (6) subjective standard; (7) perceived behavioral control; (8) willingness to follow; (9) actual behavior; (10) adapted behavior; and (11) perceived effectiveness. Each question was constructed to achieve the objective of the study. Every question in each section would help this study identify the problem’s fundamental cause and devise a solution that considers everyone’s perception.

### 2.3. Statistical Analysis: Structural Equation Modeling

Structural equation modeling (SEM) is a statistical model that employs several statistical techniques that include analysis of variance, analysis of covariance, multiple regression, factor analysis, and path analysis. SEM is also known as covariance structural analysis, equation system analysis, and analysis of moment structure [32]. Some of the popular software packages for SEM include AMOS, LISREL, and EQS; in the current study, the proposed model was obtained using AMOS 22 (New York, USA) with a maximum likelihood estimation approach. The maximum likelihood approach estimates the theoretical model parameters simultaneously to develop a full estimation model [33].

The current study utilized four sets of tests that consisted of a full model fit, the goodness-of-fit indexes, an incremental model fit, and the badness-of-fit indexes to evaluate the variation between the hypothesized model and observed data. For the full model test, the *p*-value should be greater than 0.05 with a normed chi-square (χ^2^/df) value of less than 2.0; it implies no significant difference between the observed sample and SEM estimated covariance matrices. The fundamental factor in assessing the goodness-of-fit (GOF) index of the SEM model is identifying the difference in the covariance matrices [34]. Moreover, GOF, which acts as the R^2^ in linear regression analysis, utilized the goodness-of-fit index (GFI) and adjusted goodness-of-fit index (AGFI) with the values of 0.95 and 0.90 for GFI and AGFI, respectively, which were classified as well-fitting results [35]. The incremental fix index was determined through the comparative fit index (CFI), Tucker–Lewis index (TLI), and normed fit index (NFI); a good model fit has a value greater than 0.95 for all three incremental fix index measures. Lastly, root mean square error (RMSEA) was used to evaluate the badness-of-fit index. An RMSEA value smaller than 0.07 was considered a good fit result [36].

## 3. Results

Table 2 shows the summary of the questionnaire based on the different related literature. Knowledge of COVID-19 vaccination, perceived vulnerability to disease, perceived severity to disease, perception towards government response, perceived behavioral control, willingness to follow, actual behavior, adapted behavior, and perceived effectiveness had 6 questions each while the subjective standards had 7 questions, with a total of 61 questions.

Figure 1 demonstrates the initial SEM for evaluating factors affecting the perceived effectiveness of government response towards COVID-19 vaccination in Occidental Mindoro. According to the figure, two hypotheses did not have a significant relationship:Hypothesis 3: Perceived behavioral control to perceived vulnerability (*p* = 0.76);Hypothesis 5: Perceived vulnerability to subjective standard (*p* = 0.216).In addition, 11 hypotheses had a significant relationship, as follows:Hypothesis 1: Knowledge of COVID-19 vaccination and perceived vulnerability (*p* = 0.001);Hypothesis 2: Knowledge of COVID-19 vaccination and perceived severity (*p* = 0.002);Hypothesis 4: Perceived severity and perceived behavioral control (*p* = 0.001);Hypothesis 6: Perceived severity and subjective standard (*p* = 0.002);Hypothesis 7: Perceived behavioral control and willingness to follow (*p* = 0.003);Hypothesis 8: Subjective standards and willingness to follow (*p* = 0.002);Hypothesis 9: Willingness to follow and adaptive behavior (*p* = 0.001);Hypothesis 10: Willingness to follow and actual behavior (*p* = 0.003);Hypothesis 11: Actual behavior and perceived government response (*p* = 0.002);Hypothesis 12: Adaptive behavior and perceived government response (*p* = 0.025);Hypothesis 13: Government response and perceived effectiveness (*p* = 0.002).

A revised SEM was derived by removing the two latent variables that did not have a significant relationship from the initial SEM. Figure 2 shows the final SEM for the perceived effectiveness of government response towards COVID-19 vaccination in Occidental Mindoro. The proponents modified some indices to enhance the model fit based on previous studies that used the SEM approach [14].

Table 3 shows the descriptive statistic results of each indicator. The values of the CFI, IFI, and TLI were all better than the suggested value of 0.80, which signifies that the model accurately represented the observed data. Additionally, the computed GFI and AGFI values were 0.828 and 0.801, which means that the model fit was good. The computed value of RMSEA was 0.042, which was also lower than the recommended value.

Table 4 shows the Cronbach’s alpha of nine latent variables was greater than 0.7, their average variance extracted was all greater than 0.5 and their composite reliability was also greater than 0.7.

Table 5 shows that the five parameters, namely, incremental fit index, Tucker–Lewis index, comparative fit index, goodness-of-fit index, and adjusted goodness-of-fit index, were all acceptable with parameter estimates greater than 0.8, whereas mean square error was excellent with parameter estimates less than 0.07. The RMSEA value was 0.042, lower than the recommended value. Finally, the direct, indirect, and total effects are presented in Table 6.

## 4. Discussion

The current study used the Protection Motivation Theory (PMT) and the extended Theory of Planned Behavior (TPB) to assess the factors that influence the perceived effectiveness of the government’s response to COVID-19 vaccination in Occidental Mindoro. Knowledge of COVID-19 vaccination (K), perceived severity (PS), perceived behavioral control (PBC), subjective standards (SS), willingness to follow (WF), adaptive behavior (AD), actual behavior (AB), perceived government response (PGR), and perceived effectiveness (PE) were all investigated using SEM. Through an online questionnaire, a total of 400 data samples were collected.

According to the SEM, the knowledge of COVID-19 vaccination showed a significant direct effect on PS (β: 0.331, *p* = 0.002). In order to combat COVID-19, it is critical to raise public knowledge about the disease. Similarly, in the study of Alrefaei et al. (2022), they indicated that people will be more likely to adopt and apply government guidance, especially regarding the vaccination if they understand it [78]. People who do not know enough or do not know enough about the COVID-19 vaccine may still be under pressure and have doubts. Individuals’ views of varied news and information, whether positive or negative, influence their understanding, which determines their decision to be vaccinated or not. It is necessary to understand the COVID-19 vaccine to improve health promotion and minimize immunization barriers. SEM also revealed that SS had a significant negative effect on W (β: −0.027, *p* = 0.054). According to the theory of reasoned action, an individual’s willingness to follow is determined by their intention and desired outcome, not by subjective standards.

On the other hand, PBC has been shown to improve WF (β = 0. 054, *p* = 0.001). According to Workman et al. (2008), self-efficacy significantly influences a person’s ability to complete tasks. If a person believes he or she can keep up with the new environment, he or she will be able to independently follow new legislation and the changing environment of the worldwide pandemic. Furthermore, a person with certainty has the freedom to decide whether or not to get vaccinated based on their health situation [24].

In terms of perceived government response, the results showed that AD (β: 0.054; *p* = 0.192) and AB (β: 0.686; *p* = 0.001) substantially impacted PGR. In addition, Mansoor (2021) implied that the government developing policies and solutions is essential to the public’s reactions to the pandemic. This leads to the public’s trust in the government being essential for this scenario [79]. The government maintains track of civilians’ thoughts, feelings, suffering, and dealing with the circumstance and how they perceive danger, emotions, and behavior in relation to the COVID-19 epidemic that is currently in effect.

Alternatives recommended by the government included voluntary physical distance, face masks, face shields, mass testing, and school cancellations. Meanwhile, the perceived government response was found to affect PE (β: 1.002; *p* = 0.003) significantly. Individuals must voluntarily follow the government’s new policies to be effective. In addition, Nwakasi et al. (2022) revealed that gender, education, residence town, stigma, perceived threat, and government confidence are all relevant characteristics that can influence compliance with public health mandates [80]. The government should first learn about their citizens’ reactions to the pandemic and gather data to benefit everyone’s well-being. With this, the government can create policies based on citizens’ abilities to implement them.

According to the findings, we had a substantial beneficial influence on AB (β: 1.122; *p* = 0.002). If people follow the rules, such as wearing face masks outside and keeping physical distances in mind, they may see positive results such as fewer COVID-19 cases and a healthier body. Additionally, people who look to be in good health following vaccination may convince others to get vaccinated. Additionally, explaining the FDA’s vaccine approval procedure and emphasizing the pandemic’s economic toll had the most significant impact on people’s desire to take the vaccination. The FDA therapy raises the chances of increased vaccination willingness [81].

On the other hand, the SEM revealed that WF has a substantial negative influence on AD (β: −0.383; *p* = 0.002). It could imply that no other considerations impact a person’s decisions. According to the theory of planned behavior, fear, emotion, or previous experiences have no bearing on a person’s decision to engage in an activity.

Meanwhile, PBC (β: 0.106; *p* = 0.037) and SS (β: 0.057; *p* = 0.419) had substantial direct effects on perceived severity (PS). As a result, assurance aids individuals in deciding whether or not to get vaccinated and how to react to the COVID-19 vaccine’s effects. It also takes into account an individual’s perspective on the environment. If a person’s surroundings were shown to cause serious consequences or disease, he or she would imagine the same thing occurring to him or her, which would significantly impact their behavior. Individuals with low perceived severity and low government satisfaction made the least behavioral changes, while those with high perceived severity and low government satisfaction made the greatest.

Surprisingly, since the two paths connecting perceived vulnerability to subjective standard (0.216) and perceived behavioral control (0.76) were not significant because their *p*-values were greater than 0.5, the latent variable perceived vulnerability did not affect the latent variable perceived effectiveness. This was in contrast with Li et al. (2020), who asserted that perceived vulnerability directly affects perceived behavioral control through people’s engagement and compliance with COVID-19 regulations [17].

Despite the valuable contributions, the authors would like to state several limitations of the current study. First, this study was primarily focused only on one area of the Philippines. Future research might try to conduct research all over the country to see more valuable results, which might be used to see the differences in the results between the places in the country. Second, our sample was collected online through convenience sampling. Future studies might collect data face to face to avoid any bias and to obtain more valuable input from the researchers. Lastly, the researchers did not include mental health factors that might contribute to the results of the study. Future research might include several factors such as mental health during the pandemic that might give relevant information to the research community.

## 5. Conclusions

The government should start informing the public about the essential information and other information the nation needs to know about the COVID-19 vaccination to not make the public doubt that knowledge on COVID-19 vaccination has a significant effect on perceived severity (0.331), which are the factors that stand first in line before the other factors. Then, the government agencies should pass on facts about various vaccines, processes, schedules, and side effects to not lessen the percentage of individuals accepting COVID-19 vaccination because the factor perceived severity has a significant effect on perceived behavioral control (0.106). In connection with the last statement, perceived severity has a significant effect on subjective standard (0.052). Government agencies should consider the subjective standards that most citizens abide by (Department of Health, 2021). The factors perceived behavioral control and willingness to follow had a significant relationship (1.054). The current study is one of the first to study the factors that affect the perceived effectiveness of government response toward COVID-19 vaccination. Finally, this study’s integrated PMT and extended TPB may be implemented and extended to evaluate the perceived success of government response toward COVID-19 vaccination in other countries.

## Figures and Tables

**Figure 1 healthcare-10-01483-f001:**
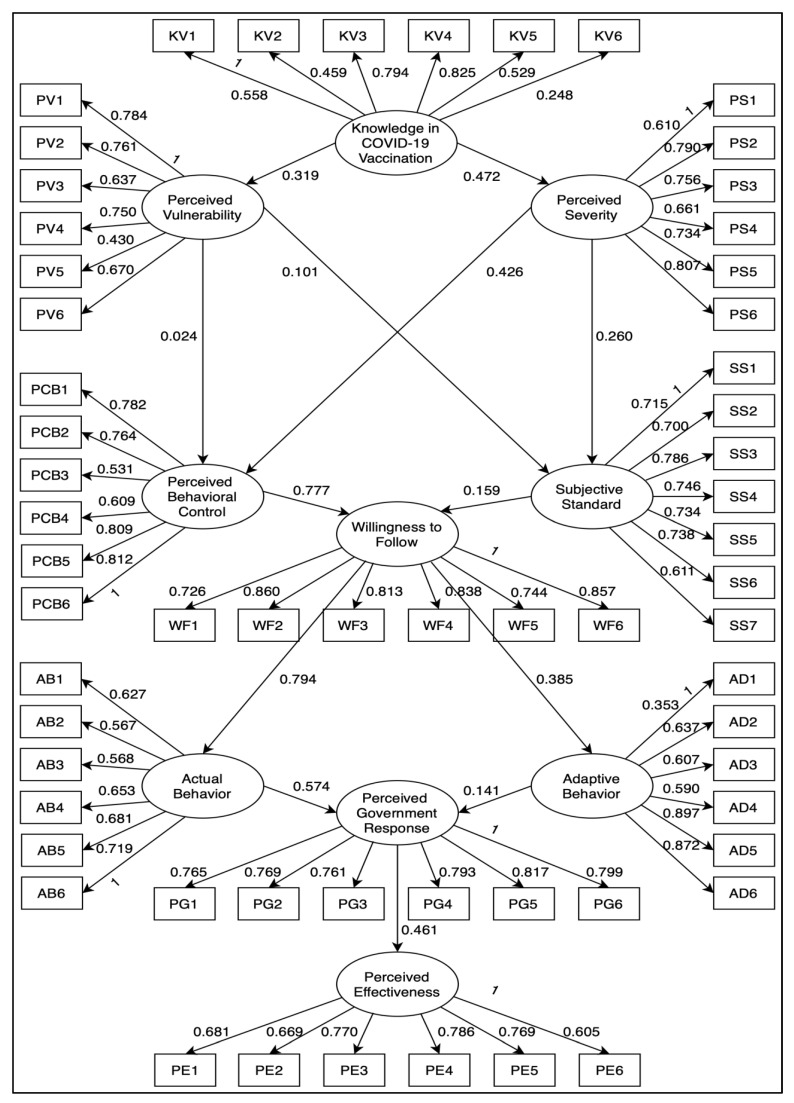
Initial SEM with indicators for evaluating factors affecting the perceived effectiveness of government response towards COVID-19 vaccination in Occidental Mindoro.

**Figure 2 healthcare-10-01483-f002:**
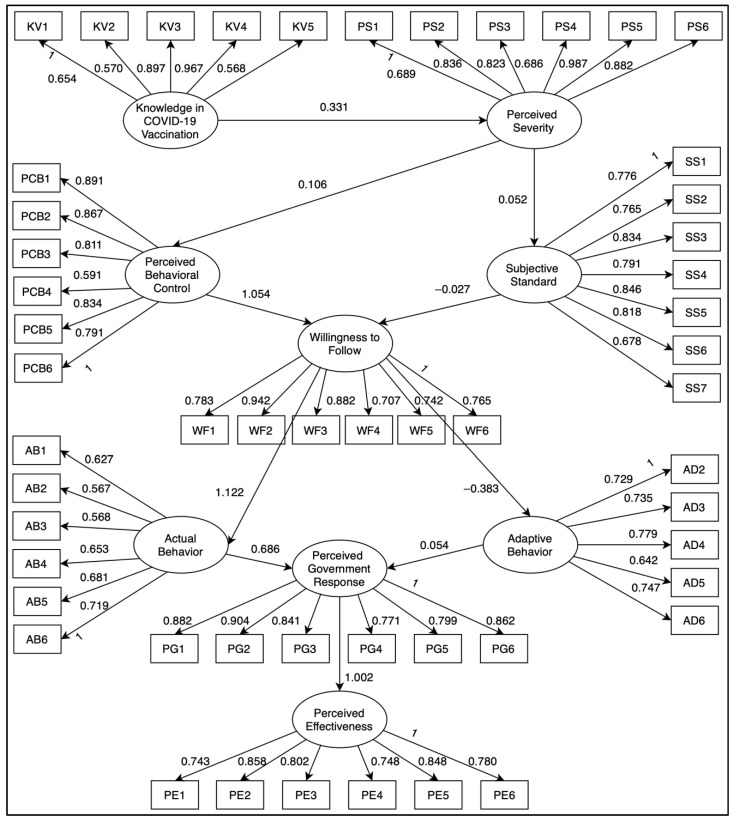
The final SEM for evaluating the perceived effectiveness of government response towards COVID-19 vaccination in Occidental Mindoro.

**Table 1 healthcare-10-01483-t001:** Demographic summary of the participants (*n* = 400).

Characteristics	Category	*n*	%
Gender	Male	173	43.2%
	Female	227	56.8%
Age	18–29	322	80.5%
	30–39	25	6.3%
	40–49	27	6.8%
	50–59	21	5.2%
	60–69	4	1%
	70 and over	1	0.2%
Educational Background	Elementary graduate	10	2.5%
	High School graduate	42	10.5%
	Senior High School graduate	168	42%
	Technical-Vocational graduate	23	5.8%
	Baccalaureate/College graduate	119	29.8%
	Post-Baccalaureate graduate	16	4%
	No grade completed	5	1.2%
	Special Education (undergraduate)	14	3.5%
	Special Education (graduate)	3	0.7%
Municipality	Abra de Ilog	18	4.5%
	Calintaan	14	3.5%
	Looc	19	4.8%
	Lubang	16	4%
	Magsaysay	50	12.5%
	Mamburao	15	3.7%
	Paluan	15	3.7%
	Rizal	50	12. 5%
	Sablayan	25	6.2%
	San Jose	141	35.2%
	Sta. Cruz	37	9.2%
Monthly salary	Less than PHP 15,000	311	77.3%
	PHP 15,001–30,000	64	16%
	PHP 30,001–45,000	20	5%
	PHP 45, 001–60,000	4	1%
	PHP 60,001–75,000	1	0.7%
	PHP above 75,000	0	0%

**Table 2 healthcare-10-01483-t002:** The constructs and measurement item.

Construct	Items	Measures	Supporting Measures
Knowledge of COVID-19 Vaccination	KV1	I think it is legally mandatory to take COVID-19 vaccines.	Kumari et al. (2021) [37]
	KV2	COVID-19 vaccination may protect other people who do not receive the vaccine.	Mohamed et al. (2021) [38]
	KV3	I think the COVID-19 vaccine will be useful in protecting me from COVID-19 infection.	Kumari et al. (2021) [37]
	KV4	I think COVID-19 vaccines are essential for us.	Islam et al. (2021) [39]
	KV5	I think COVID-19 vaccines use genetic material from coronavirus as the active ingredients.	Mohamed et al. (2021) [38]
	KV6	I think COVID-19 vaccines have health-related risk.	Alqudeimat et al. (2021) [40]
Perceived Vulnerability to Disease	PV1	I think I am highly susceptible to COVID-19.	Diaz et al. (2016) [41]
	PV2	I think there is a possibility that my family will be infected with COVID-19.	Nicola et al. (2020) [42]; Coccia (2020) [43]
	PV3	I have a history of infectious illness vulnerability.	Diaz et al. (2016) [41]
	PV4	I think I am more prone to become ill when my friends are ill.	Diaz et al. (2016) [41]
	PV5	I think COVID-19 is a major threat in my community.	Coccia (2020) [43]
	PV6	I think I am vulnerable to COVID-19 because of my job.	Bavel et al. (2020) [44]
Perceived Severity to Disease	PS1	The thought of COVID-19 gives me a negative emotion (e.g., worries, fears, and anger)	Li et al. (2020) [17]
	PS2	Contracting COVID-19 would be very serious.	Yıldırım and Güler (2020) [45]
	PS3	Thinking that I am exposed or at risk of getting COVID-19 threatens me.	Yıldırım and Güler (2020) [45]
	PS4	Contracting COVID-19 would greatly endanger my financial stability.	Shauly et al. (2020) [46]
	PS5	I believe that COVID-19 brings severe health problems.	Luo et al. (2021) [47]
	PS6	Contracting COVID-19 would threaten my family.	Stephenson et al. (2020) [48]
Perception towards Government Response	PG1	The government proactively released timely information about vaccination.	OECD (2021) [49]
	PG2	The government communicated clearly to ensure that everyone had the information they needed for the COVID-19 vaccination, regardless of socioeconomic level, migrant status, ethnicity, or language.	Lazarus et al. (2020) [50]
	PG3	The government promotes confidence in the effectiveness and safety of the vaccine.	OECD (2021) [49]
	PG4	The government had a strong COVID-19 vaccination preparedness team that included public health and medical team.	Lazarus et al. (2020) [50]
	PG5	The government made sure we always had full access to the healthcare services we needed during the COVID-19 vaccination.	Lazarus et al. (2020) [50]
	PG6	The government made certain that healthcare personnel always had the personal equipment they required to avoid contracting COVID-19.	World Health Organization (2021) [51]
Subjective Standards	SS1	Most people I know are following the preventive protocols given by the government.	Centers for Disease Control and Prevention (2020) [52]
	SS2	Most people I know are wearing face masks outside.	Rubio-Romero et al. (2020) [53]
	SS3	Most people I know are wearing face shields at enclosed public spaces (such as commercial establishments).	Parrocha (2021) [54]
	SS4	Most people I know are staying at home and/or work from home.	Barbour et al. (2021) [55]
	SS5	Most people I know are using hand sanitizer.	Mahmood et al. (2020) [56]
	SS6	Most people I know are doing physical distancing.	Guo et al. (2021) [57]
	SS7	Most people I know are vaccinated (either once or twice).	Fadda et al. (2021) [58]
Perceived Behavioral Control	PBC1	I am aware of the facts about COVID-19 vaccines and do not believe in fake news spreading in social media.	Zhang et al. (2021) [59]
	PBC2	Availability of vaccines with higher efficacy rate against COVID-19 pushed me to get vaccinated.	Zhang et al. (2021) [59]
	PBC3	It is mostly up to me if I get COVID-19 vaccine or not.	Zhang et al. (2021) [59]
	PBC4	If I wanted to, I could easily have COVID-19 vaccination.	Zhang et al. (2021) [59]
	PBC5	I believe in the effectiveness of the vaccine given by the government because it has been proven safe and effective.	Yahaghi et al. (2021) [60]
	PCB6	The availability of the vaccine here in Occidental Mindoro will push me to get vaccinated.	Yahaghi et al. (2021) [60]
Willingness to Follow	WF1	I am willing to trust in the ability of governments to communicate about vaccination.	El-Elimat et al. (2021) [61]
	WF2	I am willing to be vaccinated.	Guidry et al. (2021) [62]
	WF3	I am willing to follow the safety signal and the different responses of regulators.	Bish et al. (2011) [63]
	WF4	I am willing to coordinate with government policies during COVID-19 vaccination.	van der Bles et al. (2021) [64]
	WF5	I am willing to be vaccinated in any available vaccines in our municipality.	van der Bles et al. (2021) [64]
	WF6	I am willing to wait for my turn to be vaccinated.	Wang et al. (2021) [65]
Actual Behavior	AB1	Majority of the people are getting vaccinated.	Reiter et al. (2021) [66]
	AB2	COVID-19 vaccination is near in my area.	Reiter et al. (2021) [66]
	AB3	The company/school where I work/study implements work from home to prevent the spread of COVID-19.	Chi et al. (2021) [67]
	AB4	COVID-19 vaccine has no payment.	Kitro et al. (2021) [68]
	AB5	I am practicing social distancing to prevent the risk of spreading the virus.	Wu and Mcgoogan (2020) [69]
	AB6	I always wear face mask whenever I go.	Shaw et al. (2020) [70]
Adapted Behavior	AD1	I will wait for others to be vaccinated.	Cerda and Garcia (2021) [71]
	AD2	I do not get vaccinated because of fear of needles.	Alle and Oumer (2021) [72]
	AD3	I was worried about side effects of COVID-19 vaccine.	Reno et al. (2021) [73]
	AD4	I am worried about the rapidity of the development of the COVID vaccine.	Alle and Oumer (2021) [72]
	AD5	I think vaccine will be ineffective.	Cerda and Garcia (2021) [71]
	AD6	I think the COVID-19 vaccine was not safe.	Reno et al. (2021) [73]
Perceived Effectiveness	PE1	I think vaccine prevents me from being infected.	Mohamed et al. (2021) [38]
	PE2	I think I can lead a normal life after I get vaccinated.	Mohamed et al. (2021) [38]
	PE3	I think vaccination decreases my chance of getting COVID-19 or its effects.	Lin et al. (2020) [74]
	PE4	I think vaccines mimic the virus or bacterium that causes illness and cause the body to produce antibodies in response.	Khorramdelazad et al. (2021) [75]
	PE5	I think COVID-19 vaccines vary as does the way they stimulate the immune system to produce antibodies.	Romero-Alvarez et al. (2021) [76]
	PE6	I think my age gives an impact on how I react to the COVID-19 vaccine’s negative effects.	Djanas et al. (2021) [77]

**Table 3 healthcare-10-01483-t003:** Descriptive statistics result.

Variable	Item	Mean	Std	Factor Loading	
				Initial	Final
Knowledge of COVID-19 Vaccination	KV1	3.74	1.164	0.572	0.654
	KV2	3.43	1.234	0.460	0.570
	KV3	4.11	0.899	0.800	0.897
	KV4	4.13	0.915	0.834	0.967
	KV5	3.91	0.943	0.500	0.568
Perceived Severity	PS1	3.93	1.042	0.610	0.689
	PS2	4.25	0.829	0.790	0.836
	PS3	4.01	0.950	0.756	823
	PS4	4.05	0.946	0.661	0.686
	PS5	4.31	0.902	0.742	0.987
	PS6	4.25	0.884	0.807	0.882
Perceived Behavioral Control	PBC1	4.14	0.888	0.812	0.891
	PBC2	3.98	0.940	0.810	0.867
	PBC3	4.21	0.918	0.610	0.811
	PBC4	3.96	1.017	0.532	0.591
	PBC5	3.92	0.952	0.764	0.834
	PBC6	3.90	1.021	0.783	0.791
Subjective Standards	SS1	3.70	1.003	0.716	0.776
	SS2	4.13	0.930	0.704	0.765
	SS3	3.73	1.045	0.787	0.834
	SS4	3.79	1.002	0.748	0.791
	SS5	3.97	0.957	0.736	0.846
	SS6	3.51	1.115	0.739	0.818
	SS7	3.84	1.002	0.612	0.678
Willingness to Follow	WF1	3.92	0.899	0.857	0.783
	WF2	4.20	1.010	0.744	0.942
	WF3	4.34	0.856	0.838	0.882
	WF4	4.34	0.883	0.813	0.707
	WF5	4.03	1.047	0.860	0.742
	WF6	4.24	0.935	0.728	0.765
Adaptive Behavior	AD1	3.21	1.187	0.353	-
	AD2	2.34	1.421	0.637	0.729
	AD3	3.57	1.241	0.607	0.735
	AD4	3.53	1.139	0.590	0.779
	AD5	2.71	1.225	0.897	0.642
	AD6	2.64	1.249	0.872	0.747
Actual Behavior	AB1	3.98	0.870	0.627	0.739
	AB2	3.87	1.094	0.567	0.965
	AB3	4.10	0.941	0.568	0.918
	AB4	4.51	0.795	0.653	0.648
	AB5	4.30	0.879	0.681	0.664
	AB6	4.58	0.721	0.719	0.728
Perceived Government Response	PGR1	3.88	0.895	0.765	0.882
	PGR2	3.82	0.945	0.769	0.904
	PGR3	3.89	0.923	0.761	0.841
	PGR4	3.67	0.988	0.793	0.771
	PGR5	3.63	0.990	0.817	0.799
	PGR6	3.76	0.937	0.799	0.862
Perceived Effectiveness	PE1	3.72	1.054	0.681	0.743
	PE2	3.62	0.999	0.669	0.858
	PE3	3.90	0.945	0.770	0.802
	PE4	3.85	0.922	0.786	0.748
	PE5	3.97	0.843	0.769	0.848
	PE6	3.94	0.922	0.605	0.780
Perceived Vulnerability	PV1	3.19	1.179	0.784	-
	PV2	3.46	1.284	0.761	-
	PV3	2.23	1.330	0.637	-
	PV4	2.99	1.324	0.750	-
	PV5	4.11	1.165	0.430	-
	PV6	2.93	1.372	0.670	-

**Table 4 healthcare-10-01483-t004:** Construct validity model.

Factor	Cronbach’s α	Average Variance Extracted (AVE)	Composite Reliability (CR)
Knowledge of COVID-19 Vaccination	0.754	0.563	0.860
Perceived Severity	0.871	0.680	0.930
Perceived Behavioral Control	0.865	0.646	0.915
Subjective Standards	0.882	0.622	0.920
Willingness to Follow	0.922	0.652	0.918
Adaptive Behavior	0.848	0.530	0.849
Actual Behavior	0.803	0.620	0.905
Perceived Government Response	0.907	0.713	0.937
Perceived Effectiveness	0.859	0.636	0.913

**Table 5 healthcare-10-01483-t005:** Model fit.

Good-of-Fit Measures of SEM	Parameter Estimates	Minimum Cutoff	Interpretation
Incremental Fit Index (IFI)	0.932	>0.8	Acceptable
Tucker–Lewis Index (TLI)	0.923	>0.8	Acceptable
Comparative Fit Index (CFI)	0.931	>0.8	Acceptable
Goodness-of-Fit Index (GFI)	0.828	>0.8	Acceptable
Adjusted Goodness-of-Fit Index (AGFI)	0.801	>0.8	Acceptable
Root Mean Square Error (RMSEA)	0.042	<0.07	Excellent

**Table 6 healthcare-10-01483-t006:** Direct, indirect effect, and total effects.

No.	Variable	Direct Effects	*p*-Value	Indirect Effects	*p*-Value	Total Effects	*p*-Value
1	KV-PS	0.331	0.002			0.331	0.002
2	KV-SS			0.017	0.394	0.017	0.394
3	KV-PBC			0.035	0.031	0.035	0.031
4	KV-WF			0.037	0.031	0.037	0.031
5	KV-AD			0.041	0.03	0.041	0.03
6	KV-AB			−0.014	0.024	−0.014	0.024
7	KV-PGR			0.027	0.028	0.027	0.028
8	KV-PE			0.027	0.029	0.027	0.029
9	PS-PBC	0.106	0.001			0.106	0.037
10	PS-SS	0.052	0.002			0.052	0.419
11	PS-WF			0.111	0.036	0.111	0.036
12	PS-AD			0.124	0.042	0.124	0.042
13	PS-AB			−0.042	0.033	−0.042	0.033
14	PS-PGR			0.083	0.04	0.083	0.04
15	PS-PE			0.083	0.041	0.083	0.041
16	PBC-WF	1.054	0.003			1.054	0.001
17	PBC-SS					0	0
18	PBC-AD			1.183	0.002	1.183	0.002
19	PBC-AB			−0.404	0.002	−0.404	0.002
20	PBC-PGR			0.79	0.001	0.79	0.001
21	PBC-PE			0.791	0.003	0.791	0.003
22	SS-WF	−0.027	0.002			−0.027	0.54
23	SS-AD			−0.031	0.543	−0.031	0.543
24	SS-AB			0.01	0.54	0.01	0.54
25	SS-PGR			−0.021	0.514	−0.021	0.514
26	SS-PE			−0.021	0.53	−0.021	0.53
27	WF-AD	−0.383	0.001			−0.383	0.002
28	WF-AB	1.122	0.003			1.122	0.002
29	WF-PGR			0.749	0.003	0.749	0.003
30	WF-PE			0.75	0.003	0.75	0.003
31	AD-PGR	0.054	0.025			0.054	0.192
32	AD-AB					0	0
33	AD-PE			0.055	0.180	0.055	0.18
34	AB-PGR	0.686	0.002			0.686	0.001
35	AB-PE			0.687	0.002	0.687	0.002
36	PGR-PE	1.002	0.002			1.002	0.003

## Data Availability

Not applicable.

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
