# Peer review of "Factors Affecting Perceived Effectiveness of Government Response towards COVID-19 Vaccination in Occidental Mindoro, Philippines"

_healthcare, 2022, doi:10.3390/healthcare10081483_

Round 1

Reviewer 1 Report

My congratulations this work is well constructed. The methodology is clear, the results are well explained. The discussion does not have to be simply a summary. The discussion doesn't summarize what the results of the research can somehow be exploited and how the concept of theoretical aspects. Perhaps there is a bit of confusion between the discussion and conclusion regarding how the authors approach the research topic. Arrange discussion and conclusions following a linear discourse, also possibly inserting if you have answered the hypothesis, limitations, etc. 

Author Response

Good day!

Dear Reviewer,

Thank you for your valuable inputs and time for reviewing our paper. It helps a lot  in improving our paper.

Please see the attachment for response to your comments and suggestions.

Thank you very much!

Reviewer 2 Report

Extremely interesting paper investigating the factors influencing the level of vaccination against COVID19. There were 13 hypotheses in the study. They are illustrated in the figures on pages 10 and 11.

1.       In order to improve the legibility of Figures 1 and 2, the same markings should be used. It should be used either Perceived Severity either PS. Similarly, it should be described either the Subjective Standard or the SN.

2.       Perceived Vulnerability (PV) is missing in Figure 2 and Table 6.

3.       The authors argue that the hypothesis investigating the relationship between K and PV (H1) is true. Unfortunately, the presented results do not confirm this. PV is missing in both Figure 2 and Table 6. In this respect, Table 3 is incomplete.

4.       On page 9 there is a summary of the verification of 13 hypotheses. In order to improve the readability of this part of the work, the Reviewer proposes to create a bulleted list containing a summary of the results obtained.

Author Response

(The authors gave the same response as above.)

Reviewer 3 Report

Interesting paper, it is suggested to add ethical concerns and limitations of the study. Methodology is clear and detailed. 

Results and discussion is also very detailed, it could be added and commented in the discussion some results from similar studies. 

It is suggested to add ethical concerns and limitations of the study. 

Author Response

Good day!

Dear Reviewer,

Thank you for your time and effort in reviewing our paper.

Please see the attachment for the our response.

Thank you very much!

Reviewer 4 Report

Dear Authors, your article "Factors Affecting Perceived Effectiveness of Government Response towards COVID-19 Vaccination in Occidental Mindoro, Philippines" needs to be substantially revised before being considered for possible publication in Healthcare journal.

- The abstract is very very long, too much according to the hournal's style.

- The introduction contains a relevant part of methodology.

- Table and Figures, including captions and footnotes, are redundant and not sufficiently clear: they have to be self-explaining.

- There are too much Tables and too long: are all of them necessary?

- It is highly recommended to evaluate significant differences with proper statistical analysis in the composition of your population descripted in Table 1.

- Figure 1 and 2, as they are presented now, can't be understood by a reader, as they report only numbers and letters mixed in a big graph, creating a kind of chaos. Please try to modify them in order to create self-explaining figures, otherwise move them to a supplementary material, and report the main results in the text.

- It is not possible to have results and Discussion section together. The separate discussion section has also to include a paragraph with the limitations of your study.

- the conclusions are too long: they need to be more like "bullet-points", directly related to the main new results of your study.

- Only comprehensive or systematic reviews can have a so high nuumber of references: you have more than 80, it is definitely too much for a scientific article. Please cut at least 35 references.

Best regards,

the Reviewer

Author Response

Good day!

Dear Reviewer,

Thank you for your valuable inputs and time for reviewing our paper. It helps a lot  in improving our paper.

Please see the attachment for our response to your comments and suggestions.

Thank you very much!

Round 2

Reviewer 4 Report

The manuscript has been improved, even I still thin that references are deginitely too much, as well as Tables are too much and Figures are nor cristal clear.

Best regards,

The Reviewer

Author Response

Response to Reviewer 4 Comments

The authors would like to express their sincere gratitude to the Reviewer #4 for his/her valuable comments and for reading our paper. The response to the comments are as follows.

Point 1: The manuscript has been improved, even I still think that references are definitely too much, as well as Tables are too much and Figures are nor crystal clear.

Response 1:

Good day!

Thank you Reviewer for your valuable time and good suggestions to our paper it definitely improved our paper a lot. As a response for the round two suggestions the authors would like to state the following reasons for the your comments and suggestions:

The current study used survey questionnaire with 61 questions it is usual for this kind of study to have a high number of references. It is impossible for this kind of article to have a lesser number of references due to the number of questions inside the study. Since the topic of the current research is relatively new the authors opted to have more references in adapting the questionnaires for the study since each of the questions needs have supporting measures and established reliability from the published journals.

For the tables and figures the current study utilized the Structural Equation Modeling approach it is necessary to include all the presented tables in the manuscript in order to properly explain the results of the study. For the figures it is the best possible way to present the SEM Model the constructs and observed variables needs to be presented in that way in an SEM Model.

Hoping that we answered your comments and suggestions.

Regards,

The Authors

This manuscript is a resubmission of an earlier submission. The following is a list of the peer review reports and author responses from that submission.